# In Vivo Evaluation of Fibroblast Growth Factor Receptor Inhibition in Mouse Xenograft Models of Gastrointestinal Stromal Tumor

**DOI:** 10.3390/biomedicines10051135

**Published:** 2022-05-13

**Authors:** Patrick Schöffski, Yemarshet Gebreyohannes, Thomas Van Looy, Paul Manley, Joseph D. Growney, Matthew Squires, Agnieszka Wozniak

**Affiliations:** 1Department of General Medical Oncology, University Hospitals Leuven, Herestraat 49, 3000 Leuven, Belgium; 2Research Unit Laboratory of Experimental Oncology, Department of Oncology, KU Leuven, Herestraat 49, 3000 Leuven, Belgium; yemarshet.gebreyohannes@gmail.com (Y.G.); thomasvan.looy@gmail.com (T.V.L.); agnieszka.wozniak@kuleuven.be (A.W.); 3Novartis Pharma AG, St. Johann Campus, 4002 Basel, Switzerland; paul.manley_ext@novartis.com (P.M.); jgrowney@kymeratx.com (J.D.G.); 4Novartis Pharmaceuticals Corporation, Cambridge, MA 02139, USA; matthew.squires@novartis.com

**Keywords:** gastrointestinal stromal tumor, patient-derived xenograft, fibroblast growth factor receptor, mitogen-activated protein kinase, dovitinib, infigratinib, binimetinib

## Abstract

Advanced gastrointestinal stromal tumors (GIST) are typically treated with tyrosine kinase inhibitors, and imatinib is the most commonly used standard of care in first line treatments. The use of this and other tyrosine kinase inhibitors is associated with objective tumor responses and prolongation of progression-free and overall survival, but the treatment of metastatic disease is non-curative due to the selection or acquisition of secondary mutations and the activation of alternative kinase signaling pathways, leading to resistance and disease progression after an initial response. The present preclinical study evaluated the potential use of the fibroblast growth factor receptor inhibitors infigratinib and dovitinib alone or in combination with the mitogen-activated protein kinase inhibitor binimetinib in mouse models of GIST with different sensitivity or resistance to imatinib. Patient- and cell-line-derived GIST xenografts were established by bilateral, subcutaneous transplantation of human GIST tissue in female adult *nu/nu* NMRI mice. The mice were treated with dovitinib, infigratinib, or binimetinib, either alone or in combination with imatinib. The safety of treated animals was assessed by well-being inspection and body weight measurement. Antitumor effects were assessed by caliper-based tumor measurement. H&E staining and immunohistochemistry were used for assessing anti-mitotic and pro-apoptotic activity of the experimental treatments. Western blotting was used for assessing effects of the agents on kinase signaling pathways. Anti-angiogenic activity was assessed by measuring tumor vessel density. Dovitinib was found to have antitumor efficacy in GIST xenografts characterized by different imatinib resistance patterns. Dovitinib had better efficacy than imatinib (both at standard and increased dose) and was found to be well tolerated. Dovitinib had better efficacy in a *KIT* exon 9 mutant model, highlighting a role of patient selection in clinical GIST trials with the agent. In a model with *KIT* exon 11 and 17 mutations, dovitinib induced tumor necrosis, most likely due to anti-angiogenic effects. Additive effects combining dovitinib with binimetinib were limited.

## 1. Introduction

Gastrointestinal stromal tumors (GIST) are the most common mesenchymal malignancies of the digestive system [1] and the most common sarcomas in some geographic regions [2]. These rare tumors are driven by activating mutations in the *KIT* or platelet-derived growth factor receptor alpha (*PDGFRA*) genes, which encode respective receptor tyrosine kinases (RTK). The mutations result in constitutional activation of these RTK and downstream phosphatidylinositide 3-kinase (PI3K)/AKT and RAS/RAF/mitogen-activated protein kinase (MAPK) signaling pathways, leading to uncontrolled proliferation, differentiation, survival, metabolism, and migration of the tumor cells [1].

The dependence of tumor cells on constitutively activated KIT/PDGFRA makes GIST a logical target for treatment with tyrosine kinase inhibitors (TKI). Inoperable, metastatic GIST are currently treated with oral TKI such as imatinib, sunitinib, regorafenib, and ripretinib, usually in this sequence. While these drugs have significantly improved the progression-free and overall survival of patients with advanced disease, they do not have curative potential, maybe with the exception of postoperative (adjuvant) treatment of localized, high-risk GIST with imatinib.

Over time, almost all GIST patients develop resistance to established TKI, and the median duration of disease control on therapy typically decreases progressively with every line of treatment given [1]. Resistance to TKI is multifactorial. In most patients, TKI resistance develops through selection or acquisition of secondary mutations in the KIT or PDGFRA genes, mainly in the domains responsible for TKI binding [3,4,5]. Other resistance mechanisms include the activation of alternative RTK [6] and the crosstalk between different kinases [7]. There is still a high unmet medical need to develop and test novel categories of drugs that could potentially target other RTK to overcome secondary resistance in GIST.

Fibroblast growth factor receptor (FGFR) is one of such tentative targets. The fibroblast growth factor (FGF) and FGFR complex is a ubiquitous regulator of development and adult tissue homeostasis that bridges the peri-cellular matrix and the intracellular environment. FGFs and FGFRs have been identified in the cancer vasculature and supporting stromal cells as well as in cancer cells. FGF- and FGFR-directed reagents may be useful for targeting the cancer vasculature/stroma as well as cancer cells [8]. Recently, the presence of activating mutations or gene fusions involving *FGFR1* were described in a small subset of GIST without *KIT/PDGFRA* alterations [9,10]. Furthermore, FGFR signaling has also been postulated as a mechanism of resistance to imatinib. In particular, FGF2 was found to be overexpressed in imatinib-resistant GIST cells as well as in tumor samples from imatinib-resistant GIST patients, where also a genomic gain of FGFR2 was identified [7,11]. Moreover, the interaction of FGF2 with FGFR1 and FGFR3 restored MAPK signaling during treatment with imatinib [12].

Various FGF-targeting agents have been developed, such as antisense FGF oligonucleotides, soluble FGFRs, neutralizing antibodies, peptides corresponding to FGF functional domains, toxin-conjugated anti-FGFR antibodies, and small molecules [13]. FGFR-targeting reagents may inhibit cancer growth and some of them are currently tested in preclinical and clinical studies of human cancers, including GIST.

MAPK kinase (also known as MEK, MAP2K, MAPKK) is a key enzyme which phosphorylates MAPK. MEK is a member of the MAPK signaling cascade that is activated in various tumor types. When MEK is inhibited, cell proliferation is blocked, and apoptosis is induced. MEK inhibition has been studied in a number of preclinical studies in GIST. Two clinical trials have explored MEK inhibition in the context of treatment of GIST with imatinib or with pexidartinib.

In the present preclinical study, we tested in vivo the efficacy of two FGFR inhibitors, dovitinib (Novartis, Basel, Switzerland), a pan-TKI targeting FGFRs, VGFRA, and KIT, and infigratinib (BGJ398, Novartis, Basel, Switzerland), a selective and potent FGFRs inhibitor. These compounds were also tested either in combination with the standard of care GIST agent imatinib or in combination with the MEK inhibitor binimetinib (MEK162, Novartis, Based, Switzerland). The study was performed using patient- and cell-line-derived xenograft models of GIST, well characterized in terms of the sensitivity or resistance to established TKI.

## 2. Materials and Methods

### 2.1. GIST Xenografts

GIST xenografts were established by bilateral, subcutaneous transplantation of human GIST tissue in 6–7-week-old female adult *nu/nu* NMRI mice (Janvier Laboratories) as previously described [14]. To expand the cohort of mice for each in vivo experiment to a pre-defined number of animals, a donor animal was sacrificed by pentobarbital overdose and cervical dislocation. The ex-mouse tumor sample was excised and was cut into fragments of ±10 mm^3^. One tumor piece was inserted subcutaneously on each side of the recipient mouse, anesthetized using a 3% isoflurane (Rotacher) mixture in oxygen [15]. For the current study we used two patient-derived xenograft (PDX) models (UZLX-GIST2 and UZLX-GIST9), developed from consenting patients treated in the Department of General Medical Oncology at the University Hospitals Leuven, Belgium. Furthermore, we also used a cell line-derived xenograft model that we made using the GIST48 cell line (gift from J. A. Fletcher, Brigham and Women’s Hospital, Boston). PDX models were selected based on their molecular profile that is linked with their response to standard therapy (Table 1). All of these models are extensively characterized by our group and have already been used in several in vivo studies in our laboratory, proving their stability in terms of histopathological and molecular features as well as their sensitivity pattern if treated with established TKI in vivo [16,17]. Key characteristics of the models used in this study are presented in Table 1 and Appendix A.

The xenografting of donor material was approved by the Medical Ethics Committee, University Hospitals Leuven (S53483). The in vivo work was supported by ethics approval from the Ethical Committee for Animal Experimentation, KU Leuven (P07099, P184/2012, P052/2014), and animal experiments were performed in accordance with recommendations and national and European legislation.

### 2.2. Drugs and Reagents

Imatinib mesylate, dovitinib lactate, infigratinib, and binimetinib were provided by Novartis. Imatinib (10 mg/mL) and dovitinib (6 mg/mL) were dissolved in sterile water. Infigratinib (6 mg/mL) was prepared in acetate buffer pH 4.6 and polyethylene glycol 300 (PEG300) (1:1). Binimetinib (0.7 mg/mL) was dissolved in 1% carboxymethyl cellulose/0.5% Tween 80. All solutions were administered using oral gavaging needles at a dose of 5 mL/kg.

The primary antibodies for Western blotting (WB) and immunohistochemistry (IHC) as well as secondary antibodies and visualization systems were used as previously described [17].

### 2.3. Experimental Design

For the in vivo study, a total of 67 mice were transplanted bilaterally with human GIST tissue. We used UZLX-GIST2 (n = 32, for two experiments: passage (p) 12 and 17), UZLX-GIST9 (n = 20, p.4), and GIST48 (n = 15, p.10). Mice were handled as described previously [14]. An overview of the experimental design is given in Table 2.

In the first stage of the study, we tested infigratinib, a selective FGFR1-3 inhibitor, alone and in combination with imatinib administered for two weeks in two models (UZLX-GIST2 and -GIST48). Secondly dovitinib, a multi-targeted TKI, was evaluated for three weeks in two imatinib-resistant models (UZLX-GIST2 and -GIST9). Moreover, in the former model we also assessed the combination of dovitinib with the MEK inhibitor binimetinib. The dosing, schedule, and route of administration of experimental compounds was according to Novartis recommendation, based on previously obtained results. The treatment started when tumors reached 250–300 mm^3^. Treated tumors were compared with untreated controls and/or tumors treated with imatinib alone, used as a standard of care control group.

Tumor volume was evaluated by caliper measurement at baseline and subsequently three times per week until the end of each experiment. Tumor volume was calculated as width × length × height, with length being the greatest dimension and the other two axes perpendicular to the previous one. Tumors with starting volume < 100 mm^3^ on the first day of the experiment were excluded from the volumetric analysis. Mouse body weight and general wellbeing were followed up daily. At the end of the experiment, mice were sacrificed by pentobarbital overdose and cervical dislocation and GIST xenografted tissue was preserved for histopathological and molecular evaluation, both as snap frozen in liquid nitrogen and formalin fixed material. Mice were euthanized before the end of the pre-defined treatment period in case of excessive tumor growth (sum of the left and right tumor volume was >2000 mm^3^, body weight loss of >20% from the start of the experiment, other serious symptoms due to tumor growth or potential adverse effects of the experimental drugs). Tumors collected from mice before the final day of the experiment were excluded from histopathological evaluation.

### 2.4. Histological and Biochemical Assessments

Formalin-fixed tumor specimens were embedded in paraffin and 4 µm sections were cut for hematoxylin and eosin (H&E) and IHC staining. Histologic response (HR) was assessed using H&E-stained slides using an established scoring system [3]. Mitotic and apoptotic activity were evaluated by counting mitotic and apoptotic cells on H&E-stained slides in 10 high power fields (HPF). Phospho-histone H3 (pHH3) and Ki67 staining were used to evaluate the proliferation and cleaved PARP immunostainings assessed the apoptotic activity. The anti-angiogenic activity was assessed by measuring tumor vessel density in ex-mouse tumors using CD31 immunostaining. All assessments were performed as previously described [17].

To evaluate the effect of the treatment on KIT signaling Western blotting was performed using lysates prepared from the snap-frozen tumor specimens, as published previously [16].

### 2.5. Statistics

Comparisons between the tumor volumes on day 1 *versus* later time points and between treatment groups were completed using Wilcoxon matched pair (WMP) and the Mann–Whitney U (MWU) tests, respectively. Dell Statistica 13.1 (Dell Inc., Round Rock, TX, USA) was used for statistical analyses, and a *p* value of <0.05 was considered statistically significant.

## 3. Results

### 3.1. Tumor Volume Assessment

During the experiment, the tumor volume in control animals increased significantly compared to baseline in all but one experiment (all *p* < 0.05, WMP) (Figure 1). In a two week experiment untreated UZLX-GIST2 tumors increased in size by 39%, which was not significant however expected in this slow growing model. In the imatinib-resistant model UZLX-GIST9, imatinib treatment was associated with an expected increase in tumor size over time (251% of baseline, *p* = 0.012, WMP). In other models, tumor growth stabilization was seen in a model with dose-dependent imatinib sensitivity (UZLX-GIST2; 101%, *p* = 0.72) and in models derived from cell lines GIST48, which are known to be resistant to imatinib in vitro (56%, *p* = 0.07) (Figure 1).

Infigratinib alone led to tumor volume stabilization in both models tested. The combination with imatinib caused tumor stabilization in UZLX-GIST2 (103%; *p* = 0.87) or tumor shrinkage in GIST48 (49%; *p* = 0.01), but in both models the effect was not statistically different as compared to treatment with imatinib alone (Figure 1). In contrast, dovitinib caused tumor volume stabilization in UZLX-GIST9 (90%; *p* = 0.46) and shrinkage in the UZLX-GIST2 model (45%; *p* < 0.01). Similar effects were found when dovitinib was combined with binimetinib (41%; *p* < 0.01), which as a single agent led to tumor volume increase in the UZLX-GIST2 model (187%; *p* = 0.03) (Figure 1).

Overall, the treatment was well tolerated as assessed by the evaluation of mouse body weight (Appendix A). Single animals had to be sacrificed during experiments, but this occurred in different treatment groups and was not considered attributed to the treatment itself, with the exception of the treatment with binimetinib where two out of five mice had to be sacrificed because of the body weight decrease of >20% from the start of the experiment (Appendix A), so the toxic effect of this compound could not have been excluded. These mice were included in the tumor volume and body weight analysis (Figure 1 and Appendix A), but their tumors were not included in the final histological assessment. Detailed information about animals that dropped out of the study are presented in the Appendix A.

### 3.2. Histopathological Assessment

All models used in the experiments were previously described and used for other in vivo preclinical studies [16,18]. As expected, untreated tumors from all models showed stable morphological and immunohistochemical characteristics with diffuse KIT positivity and strong DOG1 staining (Appendix A). Molecular analysis confirmed the presence of specific *KIT* mutations as detected in the patient biopsy and previous passages of the respective models.

We assessed histological features and compared changes in histological appearance between treated and untreated tumors collected at the end of each in vivo experiment. As expected, the vast majority of control tumors showed a minimal HR (grade 1 in >95% of tumors) as presented in Figure 2. Neither infigratinib alone nor in combination with imatinib induced HR in the UZLX-GIST2 model. In GIST48, 28% of tumors treated with infigratinib had a grade 2–3 response; however, the combination with imatinib led only to grade 2 HR in 14% of samples (Figure 2. On the other hand, dovitinib caused grade ≥ 2 HR in 20% of UZLX-GIST2 and in all -GIST9 tumors analyzed. The combination of dovitinib + binimetinib led to HR grade 2 or 3 in 33% of UZLX-GIST2 samples (Figure 2).

In addition to histological response, the mitotic and apoptotic activity was assessed and compared with untreated controls of the respective models. When all xenograft models were considered, control tumors showed brisk mitotic activity with an average of 52 (UZLX-GIST2), 17 (GIST48), and 37 (UZLX-GIST9) mitotic figures per 10 high power fields.

In the model with known dose-dependent imatinib resistance (UZLX-GIST2), imatinib did not change the mitotic activity. Similarly, there was no effect observed under the treatment of infigratinib both alone and in combination with imatinib (Table 3). On the contrary, dovitinib led to a >50 fold decrease in the mitotic count (*p* < 0.001), which was even more pronounced when combining dovitinib with binimetinib (*p* < 0.001), without reaching statistical difference between these two arms. The apoptotic activity was induced only in tumors treated with imatinib (*p* < 0.05); however, dovitinib both alone and in combination with binimetinib caused a significant decrease in the apoptotic count (2.4- and 3.9-fold, respectively, *p* < 0.005 for both). A decrease in microvessel density as assessed by CD31 staining was observed with dovitinib (2.0-fold; *p* < 0.005) and with binimetinib (1.2-fold; *p* < 0.05) as well as in the dovitinib/binimetinib combination (2.4-fold; *p* < 0.005).

In the GIST48 cell line-derived model, imatinib alone and in combination with infigratinib was associated with a decrease in mitotic activity as compared with control (14.8- and 23-fold, respectively; *p* < 0.005), but there was no significant difference comparing the two treatment arms. On the other hand, there was no significant difference in apoptotic activity in any of the treatment groups as assessed on H&E. Interestingly, the imatinib/infigratinib combination led to a significant decrease in microvessel density when compared with control (1.3-fold; *p* < 0.005).

In UZLX-GIST9, a TKI-resistant model, imatinib, did not have any effect on the mitotic and apoptotic levels. Interestingly, dovitinib treatment led to a slight increase in proliferation (1.4-fold, *p* < 0.05). None of the treatments caused anti-angiogenic effects when comparing with the untreated tumors. The results of mitotic and apoptotic activity in all models according to H&E staining were confirmed further using IHC markers (Table 3).

### 3.3. RTK Signaling Pathways

Western blotting was performed to assess the effect of inhibitors on RTK signaling pathways. This analysis confirmed the expression and activation of KIT and its downstream intermediates in control tumors from all tested models (Figure 3). As previously reported, imatinib inhibited KIT phosphorylation and its downstream effectors only in the GIST48 model, regardless of the presence of resistant mutations. Infigratinib alone did not influence the activation of the RTK signaling pathway neither in UZLX-GIST2 nor in GIST48. However, when combined with imatinib it showed a slight enhanced effect on phosphorylation of pathway intermediates. On the other hand, dovitinib showed a mild inhibitory effect on KIT and AKT activation in the UZLX-GIST2 model, which was more evident when combined with binimetinib, leading to the almost complete absence of phosphorylated forms of AKT and MAPK. Interestingly, in both dovitinib-based treatment groups there was a pronounced increase in KIT expression. In contrast, dovitinib led to only low level of KIT inhibition as well as slight deactivation of its signaling pathway in the TK-resistant model UZLX-GIST9 (Figure 3).

## 4. Discussion

In this preclinical study, we evaluated the antitumor efficacy of two FGFR inhibitors, dovitinib (pan-TKI) and infigratinib (anti-FGFR), in PDX models of GIST, characterized by diverse molecular background and different sensitivity to imatinib as the clinical standard of care in this rare disease.

In the clinic, imatinib has revolutionized the treatment of advanced GIST [1]; however, even patients who initially respond to this treatment develop resistance over time, which most commonly is caused by the development of secondary mutations in *KIT* [3,19]. Most of these acquired mutations occur in the ATP/drug binding pocket (exons 13 and 14) or in the activation loop (exon 17) [5,20]. Although the approved second- and third-line agents (sunitinib and regorafenib) show activity against some of these resistant *KIT* mutations, the duration of disease control achieved with these agents tends to be much shorter than what can be achieved in first line treatment with imatinib in typical patients [1]. Because of the heterogeneity of acquired mutations, leading to the presence of the multiple alterations in one patient or even in one organ/one metastatic lesion [5], more specific KIT inhibitors have only a limited effectiveness in highly resistant GIST. Moreover, it was also hypothesized that therapeutic KIT inhibition may in turn activate alternative RTK such as AXL, MET, and/or FGFR1/3 [6,7,21].

For this reason, GIST patients, especially those with refractory tumors, may benefit from therapy with multi-target TKI such as dovitinib. This agent is an orally active small molecule that exhibits potent inhibitory activity against several RTKs, e.g., KIT, VEGRF1-3, FGFR1-2, and PDGFRA [22]. In GIST cell lines, dovitinib inhibits cell proliferation, although imatinib still appeared to be more potent in cell lines with varying imatinib-sensitivity. Moreover, in a xenograft model, derived from the imatinib-sensitive GIST T1 cell line, dovitinib decreased the tumor volume in a similar way as imatinib [23].

In our dose-dependent sensitive model (UZLX-GIST2), which is characterized by a *KIT* exon 9 mutation, dovitinib indeed led to tumor shrinkage (to 45% of baseline). In this model, a standard dose of imatinib (50 mg/kg/BID) was previously found to be insufficient to gain tumor growth control [17]. The effect of dovitinib on the tumor volume in our study was most likely attributed to the inhibition of KIT which resulted in an almost complete absence of proliferation. Interestingly, the apoptotic activity was significantly lower in the dovitinib-treated tumors in comparison to the control. Similar effects have been observed when this model was treated with other multi-kinase inhibitors [17]. This could be due to specific KIT genotype dependent differences in expression and activation of proteins involved in the KIT signaling pathway [24]. On the other hand, in the imatinib resistant model (UZLX-GIST9) with double *KIT* exon 11 and 17 mutations, we observed tumor volume stabilization in the dovitinib-treated group. This observation should not be underestimated as in refractory GIST tumor growth delay, resulting from a given treatment, can be considered beneficial as it prolongs the time to progression and may improve patient survival, as has been shown in the clinic [1]. On the histological level, dovitinib induced pronounced HR (grade 2 and 3) in UZLX-GIST9, while all untreated controls and >90% of tumors treated with imatinib had only HR grade 1. Interestingly, we did not see any effect of dovitinib on cell proliferation or on apoptotic activity in this model. This efficacy of dovitinib in vivo is in line with clinical results from two phase 2 trials [25,26], testing dovitinib in imatinib/sunitinib-resistant GIST patients. On the molecular level, there was no clear correlation between the presence or absence of secondary mutations, when genotyping circulating tumor DNA in serum, and the activity of dovitinib. However, none of the patients who achieved disease control at 24 weeks were found to have a *KIT* exon 17 mutation detected in serum. Moreover, patients with acquired mutations had a shorter progression free and overall survival [27]. These observations might suggest that RTKs other than mutated KIT were inhibited by dovitinib in these patients. Interestingly, in our experimental setting the model with double KIT exon 11 + 17 responded to treatment with dovitinib.

Inhibitors such as infigratinib were designed to specifically and selectively act on the tyrosine kinase domain of FGFR family members, counteracting their phosphorylation at nanomolar concentrations. Infigratinib has been approved by regulatory agencies for use in adults with previously treated, unresectable, locally advanced or metastatic cholangiocarcinoma with a FGFR2 fusion or other alterations [28] and was also evaluated clinically in other solid tumors with FGFR alterations [13]. None of our models used for in vivo evaluation showed any FGFR aberration or mutation. In GIST it was shown that FGFR signaling pathway activation could be responsible for resistance to imatinib [7,11]. Furthermore, infigratinib enhanced the growth inhibition of GIST cells caused by imatinib [12]. However, in our in vivo experiments we could not observe any anti-tumor effect with infigratinib, but we did not detect any FGFR alteration or abnormal expression in either of the xenograft models used.

We also evaluated a combination of FGFR inhibitors with different TKI. First, the combination of dovitinib with MEK inhibitor binimetinib was evaluated in the *KIT* exon 9 UZLX-GIST2 model. It was previously shown that KIT inhibition by imatinib frequently results in a rebound in MAPK phosphorylation, which could be due to a feedback activation of FGFR signaling [12]. Bauer et al. observed that MEK inhibition with U0126 completely inhibited MAPK phosphorylation in GIST cell lines in vitro and a moderate anti-proliferative effect was seen in imatinib-sensitive GIST882 cells, but only minor effects were visible in imatinib-resistant GIST lines [29]. We showed that MEK inhibition alone did not produce any significant effect on tumor volume, cell proliferation, or apoptosis. Only a moderate effect on vessel density was observed, where the anti-angiogenic effect was better than in untreated tumors. The ineffectiveness of MEK inhibition alone was also confirmed clinically in patients with advanced GIST [30]. Treatment with sunitinib plus MEK inhibitor PD-0325901 was effective in a renal cell carcinoma PDX model. The addition of MEK inhibitor abrogated resistance and led to improved anti-tumor efficacy [31]. Combined treatment of gefitinib with MEK inhibitors was shown to be therapeutically useful in lung adenocarcinoma cells with acquired gefitinib resistance and EGFR mutations [12,32].

Furthermore, we also tested the potential additive effect of the combination of infigratinib with imatinib. There, even though we could see the complete inactivation of the downstream signaling pathway in UZLX-GIST2 (with *KIT* exon 9 mutation) or in GIST48, this effect did not translate into an additive effect on proliferation inhibition and tumor growth decrease that would be more pronounced than the one achieved with imatinib alone. Recent studies had proven that inhibition of FGF signaling in imatinib-resistant GIST restored the sensitivity to imatinib both in vivo and in vitro [11], though we could not confirm this in our setup. Interestingly it was also observed that inhibition of FGFR signaling in imatinib-resistant GIST cells sensitizes them to DNA-damaging agents, such as topoisomerase II inhibitors. Importantly, when FGFR2 was knocked down by small interfering RNA and/or inhibited with infigratinib, it led to decreased expression of RAD51 after doxorubicin exposure in imatinib-resistant GIST cells suggesting the attenuation of DNA repair mechanisms, providing the potential mechanism of GIST sensitization to DNA damaging agents [11,33]. This hypothesis would require further evaluation in future preclinical and clinical studies. Of note, a phase Ib study with infigratinib together with imatinib in patients with advanced GIST was stopped prematurely after inclusion of only 12 evaluable patients due to toxicity of the combination and without defining a recommended phase II dose [34].

## 5. Conclusions

Our results show that dovitinib has potential antitumor efficacy in GIST xenograft models characterized by different mechanisms of resistance. In both models tested, dovitinib showed better efficacy than imatinib (both at standard and higher doses) and was found to be well tolerated. Overall, dovitinib had better efficacy in the tested *KIT* exon 9 mutant model, suggesting a role of this genotype as a marker for patient selection. The efficacy of dovitinib in UZLX-GIST9 (with *KIT* exon 11 + 17 mutations) was observed mainly through induction of necrosis which might be secondary to the anti-angiogenic effects of this compound. Enhanced effects of a combination of dovitinib with binimetinib in UZLX-GIST2 was only visible on the KIT signaling.

At the same time, infigratinib alone did not show any in vivo anti-tumor effect in models tested and the combination with imatinib did not lead to better results than imatinib alone.

## Figures and Tables

**Figure 1 biomedicines-10-01135-f001:**
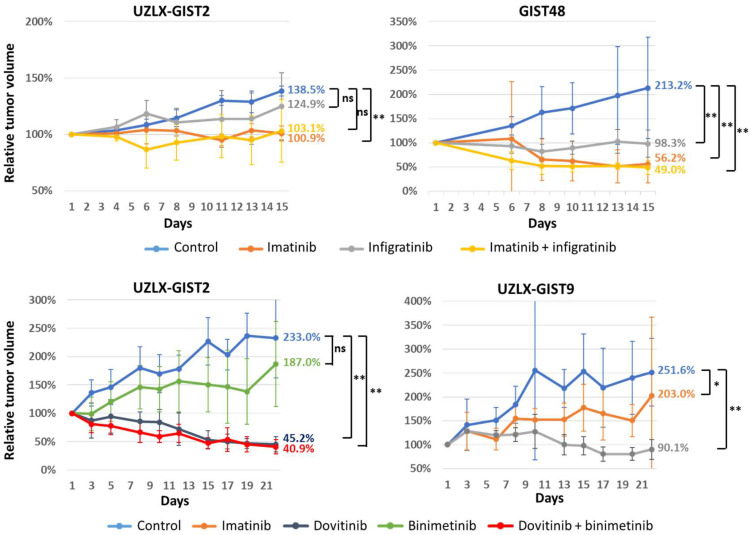
Evolution of tumor volume. Legend: Relative tumor volume evolution in the experiment testing infigratinib for two weeks and dovitinib for three weeks. The average values per were compared with control groups in the respective experiments, Mann–Whitney U test was performed to assess the difference between treatment versus respective control groups: ns—not statistically significant; * *p* < 0.05; ** *p* < 0.005.

**Figure 2 biomedicines-10-01135-f002:**
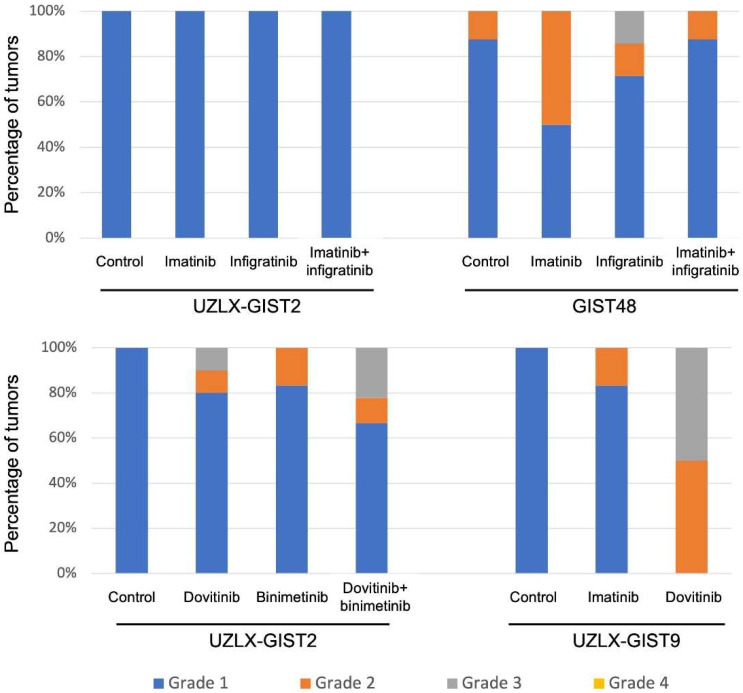
Histological response. Legend: Assessment of histological response graded by assessing the magnitude of necrosis, myxoid degeneration, and/or fibrosis on H&E staining: grade 1 (0–10%), grade 2 (>10% and ≤50%), 3 (>50% and ≤90%), and grade 4 (>90%) [3].

**Figure 3 biomedicines-10-01135-f003:**
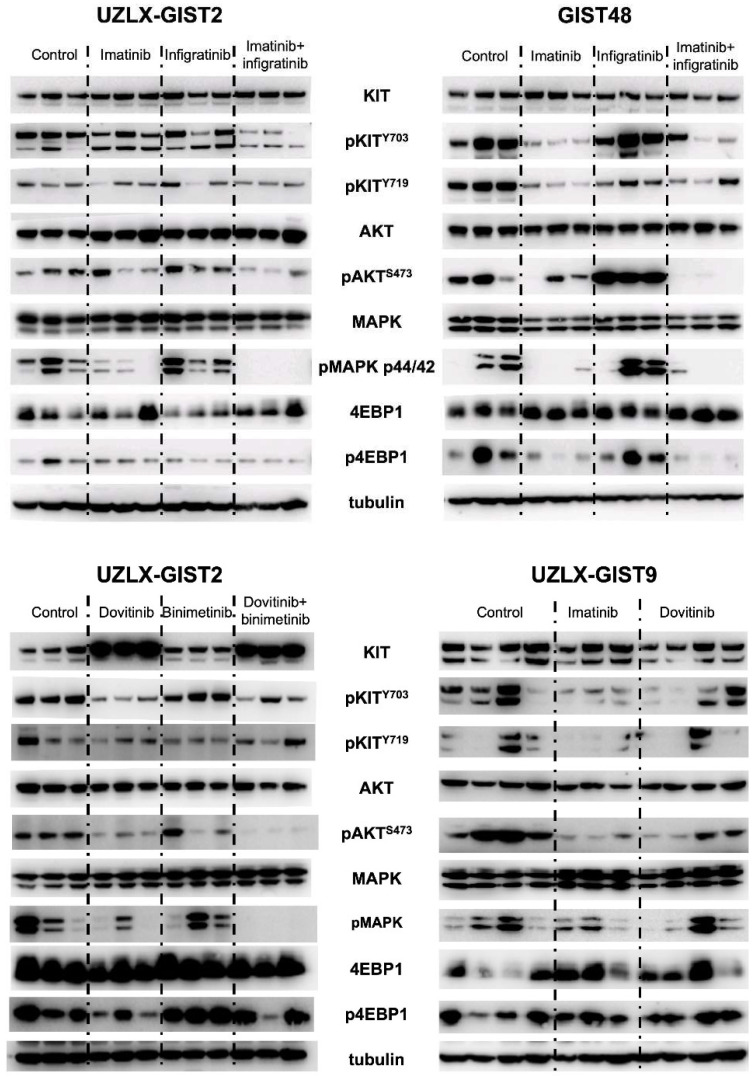
Western blotting results. Legend: Analysis of receptor tyrosine kinase signaling pathway for tumors collected after treatment, using three biological replicates per treatment group.

**Table 1 biomedicines-10-01135-t001:** Characterization of xenograft models used for the experimental work.

Xenograft Model		Histopathological Characteristics	*KIT* Mutation	In Vivo Sensitivity to Standard TKI
UZLX-GIST2	Patient-derived	Spindle cells KIT(+), DOG1(+)	p.A502_Y503dup	Imatinib dose-dependent sensitive Sunitinib sensitive
UZLX-GIST9	Patient-derived	Spindle cells KIT(+), DOG1(+)	p.P577del;W557LfsX5;D820G	Imatinib resistant Sunitinib resistant
GIST48	Cell line-derived	Spindle cells KIT(+), DOG1(+)	p.V560D;D820A	Imatinib sensitive Sunitinib sensitive

Legend: (+) immunopositivity; DOG1—discovered on GIST 1; TKI—tyrosine kinase inhibitor.

**Table 2 biomedicines-10-01135-t002:** Experimental set-up including model/passage used, treatment dose, and schedule. The number of mice represents animals that entered the experiment, the number of tumors denote samples taken into the account for the final analysis.

Model Name	Passage	Number of Mice/Tumors Per Treatment Group						
		Control (Vehicle)	Imatinib (50 mg/kg BID)	Dovitinib (30 mg/kg QD)	Binimetinib (3.5 mg/kg BID)	Dovitinib + Binimetinib *	Infigratinib (30 mg/kg QD)	Imatinib + Infigratinib *
**UZLX-GIST2**	12	2/4	2/4	n/a	n/a	n/a	3/6	4/7
	17	6/12	n/a	5/10	5/6	5/9	n/a	n/a
**UZLX-GIST9**	4	7/8	6/6	7/7	n/a	n/a	n/a	n/a
**GIST48**	10	4/8	3/6	n/a	n/a	n/a	4/7	4/8

Legend: BID—bi-daily; QD—daily; n/a—not applicable; * for combination treatment doses and schedules were as in the single-treatment arms.

**Table 3 biomedicines-10-01135-t003:** Histological assessment of proliferative and apoptotic activity, performed on tumors collected after the treatment. Results are shown as fold changes in comparison with control.

		Mitotic and Proliferative Activity						Apoptotic Activity				Microvessel Density	
Xenograft Model	Treatment Group	H&E		pHH3		Ki67		H&E		Cleaved PARP		CD31	
UZLX-GIST2													
	Imatinib	=		=		=		↑ 1.8	*	=		=	
	Infigratinib	=		=		=		=		=		=	
	Imatinib+infigratinib	=		=		↓ 1.8	*	=		=		=	
	Dovitinib	↓↓↓	**	↓ 44.8	**	↓↓↓	**	↓ 2.4	**	↓ 5.3	**	↓ 2.0	**
	Binimetinib	=		=		↓ 1.9	**	=		=		↓ 1.2	*
	Dovitinib+binimetinib	↓↓↓	**	↓↓↓	**	↓↓↓	**	↓ 3.9	**	↓ 5.4	**	↓ 2.4	**
GIST48													
	Imatinib	↓ 14.8	**	↓ 7.1	**	↓ 12.1	**	=		=		=	
	Infigratinib	=		=		↓ 1.3	*	=		↑ 1.4	*	=	
	Imatinib+infigratinib	↓ 23.0	**	↓ 13.5	**	↓↓↓	**	=		=		↓ 1.3	**
UZLX-GIST9													
	Imatinib	=		=		=		=		=		=	
	Dovitinib	↑ 1.4	*	↑ 1.5	*	↑ 1.3	*	=		=		=	

Legend: =—no significant change; ↑—increase; ↓—decrease; ↓↓↓—>50-fold decrease; * *p* < 0.05; ** *p* < 0.005; H&E—hematoxylin and eosin staining; PARP—poly (ADP-ribose) polymerase; pHH3—phospho-histone H3.

## Data Availability

Not applicable.

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
