# Peer review of "In Vivo Evaluation of Fibroblast Growth Factor Receptor Inhibition in Mouse Xenograft Models of Gastrointestinal Stromal Tumor"

_biomedicines, 2022, doi:10.3390/biomedicines10051135_

Round 1

Reviewer 1 Report

GIST are very rare malignant tumors (albeit one of the most common sarcoma subtypes).  The most effective drug for treating patients with most types of GIST´s is imatinib. Unfortunately, in patients with imatinib-refractory GIST, disease control can often only be achieved for a very limited time. Due to the limited therapeutic options for patients with advanced GIST, further treatment options are urgently needed.

In the submitted manuscript the authors present a preclinical study evaluating the potential use of the fibroblast growth factor receptor inhibitors infigratinib and dovitinib alone or in combination with the mitogen-activated protein kinase inhibitor binimetinib in mouse models of GIST with different sensitivity or resistance to imatinib. They describe precisely and explain very well the molecular rationale for the chosen interventions and experiments.

All experiments are explained very well. The results are also clearly presented, well explained and adequately discussed.

Overall, this is a very well-written, easy-to-understand manuscript on extensive preclinical investigations into a very interesting drug intervention in a disease with a high need for new treatment options, i.e. with a very relevant question.

However, a small revision could be considered: Only the results of the investigations with dovitinib are summarized in the Abstract and in the Conclusions of the manuscript. However, since not only the substance dovitinib but also infigratinib alone and in combination with other substances was tested in the course of the investigations carried out, it would it make sense to also briefly mention the summary of these investigations both in the Abstract and in the Conclusions.

Reviewer 2 Report

Schöffski and co-authors present a manuscript focusing on a preclinical xenograft model of GIST and treatment response to different targeted therapies such as FGFR-Inhibitors and MEK-Inhibitors for Imatinib sensitive and resistant GIST.

The concept of preclinical modelling and treatment response is intriguing, especially if pdx-models are used. The advantages (and disadvantages) of pdx-models have been described previously (e.g. Kopetz et al, Clinical Cancer Research 2014). A study, that gains its strength from pdx should introduce these benefits.

The methodology of the work is not described sufficiently and the papers (self) quoted do not explain the methods either. The novelty of the work is unclear, especially as some mechanisms have been described including xenograft models in 2015 (Li et al, Cancer Discovery), and some drugs are currently tested in clinical settings (Chi et al, JCO, 2022). A preclinical pdx-trial would have to include by far a larger number of models than the 2 offered here. I strongly recommend the authors to consult works regarding design of pdx-studies from authros like Trussolini, Bardelli, Bertotti or Dienstmann.

Besides that, there are several open questions regarding the experimental design and the scientific significance of this work.

Major issues:

  • The authors have used 2 pdx models and one cell-line xenograft. The rational of combining these is unclear. A “preclinical” study close to the patient may focus on pdx-models, the reason for cell-line xenografts remains unclear.
  • Using only two models does not allow any conclusions on in regard to translational expectations (lines 422 ff).
  • The diagnostic procedures are not described adequately and the quoted paper in line 107 refers to another paper which in turn refers again to another paper – and this by no means offers reproducibility. How were the tissues implanted? Did the authors use Matrigel? How was tumor-volume calculated? At which size did treatement commence? Or did it commence with implantation? Were both tumors for one animal included no matter which size they had? Was there an upper limit for tumor volume?
  • Even though the authors claim their tumors to be well characterized, the self quoted works (lines 114 ff) are from 2014 and 2016, respectively. The tumors used have been passaged 12 and 17 times, respectively (were these tumors thawed and implanted? Or have they been implanted throughout?). Features close to the patient diminish over time in pdx-models. Furthermore STR-analysis for identification of these tumors is essential.
  • Is there any literature for dosages used? They are not the same as for humans.
  • What exactly do the authors mean by “model” (e.g. Table 2) ? Is this the number of actual tumors, of mice or of pdx-models?
  • Are there any statistical thoughts on why 2 and 3 models respectively for UZLX-GIST2 are enough to draw any conclusions? Drawing conclusions from two tumors of unknown size in regard to effects of treatment in a pdx-model is unscientific!
  • In addition to the previous point presenting results with relative values may be sound, but not for two tumors. Furthermore no standard deviation or standard error is seen and offered.
  • 6 of 67 animals had relevant body weight loss. The conclusion that treatment was well tolerated if almost 10% of animals clearly suffered is bold.
  • What exactly does the Western-Blot analysis show? Are we seeing biological or technical triplicates? Why are there so many fundamental discrepancies in the bands among triplicates (e.g. pMAPK)

Minor issues:

The conclusion part needs reviewing in regard to English language
